# Rabbit and Human Angiotensin-Converting Enzyme-2: Structure and Electric Properties

**DOI:** 10.3390/ijms252212393

**Published:** 2024-11-19

**Authors:** Svetlana H. Hristova, Trifon T. Popov, Alexandar M. Zhivkov

**Affiliations:** 1Department of Medical Physics and Biophysics, Medical Faculty, Medical University—Sofia, Zdrave Str. 2, 1431 Sofia, Bulgaria; 2Medical Faculty, Medical University—Sofia, Zdrave Str. 2, 1431 Sofia, Bulgaria; 3Scientific Research Center, “St. Kliment Ohridski” Sofia University, 8 Dragan Tsankov Blvd., 1164 Sofia, Bulgaria

**Keywords:** proteins, enzymes, angiotensin-converting enzyme-2 (ACE2), macromolecular models, 3D structural stability, folding energy, protein electrostatics, isoelectric point, surface electrostatic potential

## Abstract

The angiotensin-converting enzyme-2 (ACE2) is a transmembrane glycoprotein, consisting of two segments: a large carboxypeptidase catalytic domain and a small transmembrane collectrin-like segment. This protein plays an essential role in blood pressure regulation, transforming the peptides angiotensin-I and angiotensin-II (vasoconstrictors) into angiotensin-1-9 and angiotensin-1-7 (vasodilators). During the COVID-19 pandemic, ACE2 became best known as the receptor of the S-protein of SARS-CoV-2 coronavirus. The purpose of the following research is to reconstruct the 3D structure of the catalytic domain of the rabbit enzyme rACE2 using its primary amino acid sequence, and then to compare it with the human analog hACE2. For this purpose, we have calculated the electric properties and thermodynamic stability of the two protein globules employing computer programs for protein electrostatics. The analysis of the amino acid content and sequence demonstrates an 85% identity between the two polypeptide chains. The 3D alignment of the catalytic domains of the two enzymes shows coincidence of the α-helix segments, and a small difference in two unstructured segments of the chain. The electric charge of the catalytic domain of rACE2, determined by 70 positively chargeable amino acid residues, 114 negatively chargeable ones, and two positive charges of the Zn^2+^ atom in the active center exceeds that of hACE2 by one positively and four negatively chargeable groups; however, in 3D conformation, their isoelectric points pI 5.21 coincide. The surface electrostatic potential is similarly distributed on the surface of the two catalytic globules, but it strongly depends on the pH of the extracellular medium: it is almost positive at pH 5.0 but strongly negative at pH 7.4. The pH dependence of the electrostatic component of the free energy discloses that the 3D structure of the two enzymes is maximally stable at pH 6.5. The high similarity in the 3D structure, as well as in the electrostatic and thermodynamic properties, suggests that rabbit can be successfully used as an animal model to study blood pressure regulation and coronavirus infection, and the results can be extrapolated to humans.

## 1. Introduction

The angiotensin-converting enzyme-2 (ACE2) is an integral membrane glycoprotein, expressed on the extracellular surface of the cytoplasmic membranes of human and animal epithelial cells. The main function of ACE2 as a carboxypeptidase (exopeptidase) is the reduction of blood pressure by detaching one amino acid residue from the C-end of angiotensin-1 (inactive decapeptide) and its derivative, angiotensin-2 (octapeptide, the strongest vascular constrictor) (Figure 1) [1,2,3,4]. ACE2 was identified as an isoform of the ACE enzyme in 2000 [5,6,7,8]; before then, researchers did not distinguish these two enzymes, considering them identical. At present, ACE2 is best known as a receptor for the S-protein of the β-coronavirus SARS-CoV-2 (originator of the COVID-19 pandemic), although this role can also be played by other membrane-integrated proteins [9,10,11,12]. ACE2 is highly expressed in epithelial cells of the small intestine and colon, kidney, gallbladder, testis, blood vessels (endothelium), as well as in cardiomyocytes, placental trophoblasts, etc.; in the respiratory tract, ACE2 expression is not so high [13,14,15,16].

ACE2 is a conservative protein, since the analysis of its primary amino acid sequence reveals relatively small changes during evolution: the identity is 83%, 81%, 83%, 61%, 60%, and 59% for civet, bat, bird, snake, frog, and fish, respectively, as compared with the human polypeptide chain [14]. The human ACE2 protein (hACE2) is encoded by a gene, located on the short arm of the X chromosome, which consists of 18 exons and 22 introns. The full-length hACE2 polypeptide chain contains 805 amino acids residues, which consists of a signal peptide on the extracellular N-terminus (1–19 residues), a large hydrophilic carboxypeptidase segment (20–615 residues), and a smaller hydrophobic intramembrane collectrin-like segment (616–805 residues). The molecular mass of hACE2 is 92.5 kDa of the whole polypeptide chain, 90.7 kDa without the signal peptide, and 69.1 kDa of the catalytic domain, all calculated according to the amino acid content without the covalently bound saccharides molecules in the real ACE2 glycoprotein. ACE2 in its final form is a membrane-integrated globular protein consisting of a large hydrophilic catalytic domain exposed to the extracellular aqueous medium and a small hydrophobic nonenzymatic domain integrated into the cytoplasmic membrane. In contrast to the ACE isoform, the ACE2 protein has only one catalytic domain (from 147 to 555 amino acid residues, numbered from the N-end), which shares 42% identity with each of the two ACE catalytic domains [6].

The ACE2 catalytic domain has a single metalloproteinase active site, constructed by the zinc-binding motif His-Glu-Met-Gly-His (HEMGH). The amino acid residues, which form the active site of this enzyme, are hydrophilic and charged positively (His-374, His-378, His-505), negatively (Glu-375, Glu-402, Glu-406), and uncharged (Tyr-515). One divalent Zn^2+^ cation is coordinately bound to five atoms: nitrogen of the imidazole groups of the two histidine residues from the zinc-binding motif (His-374 and His-378), one oxygen atom of the glutamic acid from the zinc-binding motif (Glu 375), and the two oxygen atoms of the ionized carboxylic group (–CO^−1/2^O^−1/2^) of the amino acid residue of the C-terminus of the substrate (leucine residue in the case of angiotensin-1) [17]. The enzyme splits the peptide bond between the first and second amino acid residues (on the C-end of the peptide chain) due to the shift of the π-electron density in the side carboxylic group –C=O to the zinc cation in the coordination bond –C=O…Zn^2+^, which results in the weakening of the peptide bond (–NH–CO–). The catalytic center is located around a pocket in the extramembrane domain; its size and shape of the pocket determine different enzymatic activity [17,18,19,20].

Besides the hydrophobicity of the C-end residue, the enzyme activity probably depends on the electric charges of the angiotensin peptides and the electrostatic potential on the surface of the catalytic domain of the ACE2 enzyme because the adsorption of the peptide is a preliminary step in the enzyme reaction. The decapeptide angiotensin-I has four positively charged groups: one arginine residue, two histidine residues, and –NH_3_^+^ group of the N-end aspartic acid, and two negatively charged groups: one of the N-end aspartic acid and the deprotonated carboxylic group of the C-end leucine. The octapeptide angiotensin-II has three positive charges (arginine, histidine, and N-end aspartic acid) and two negative charges (N-end aspartic acid and the COO^−^ group of the C-end phenylalanine) (Figure 1).

In this article, we compare the rabbit (rACE2) and the human (hACE2) angiotensin- converting enzyme-2 by considering their amino acid sequence, 3D structure, pH-dependent thermodynamic stability, hydrophilicity/hydrophobicity, and the local electrostatic potential on the surface of the catalytic domain. The comparison of hACE2 and rACE2 follows the need to extrapolate the experimental results obtained for the rabbit to the human ACE2. Although the catalytic site is conservative—in particular, it is identical in the hACE2 and rACE2—there is some difference in the structural stability and the electric properties because they are determined by the whole polypeptide chain and its 3D conformation. The three-dimensional (3D) structure of the human ACE2 is determined by crystallographic X-ray analysis of the crystallized hydrophilic extramembrane domain. The 1R42 model of hACE2, deposited in the Protein Data Bank (PDB), has 597 amino acid residues (from 19 to 615, numbered from the N-end) with molecular mass 76.77 kDa. In the PDB, there is no deposited 3D model of a free rabbit ACE2; only its primary amino acid sequence is published. Therefore, before calculation of the thermodynamic and electrostatic properties of rACE2, it is necessary to reconstruct its 3D structure. To do that, we replace the different amino acid residues in the 3D model of the human ACE2 with the corresponding residues in the rabbit ACE2, applying mutation analyses by a computer program as we did for single mutations in our previous research [21].

Our interest in the rACE2 is conditioned by the use of rabbits (mainly European rabbit—Oryctolagus cuniculus) as experimental animals, along with rats, mice, guinea pigs, and bulls [22,23,24,25]. Compared to rodents, rabbit genes encoding important components of the immune system are phylogenetically more similar to human genes, suggesting a greater similarity of immune response in studies of various infectious diseases between rabbit and human than mouse and human, for example. There are numerous studies in which rabbits are used as model animals for the investigation of both infectious (tuberculosis, syphilis, human immunodeficiency virus—HIV, human papillomaviruses, etc.) and non-infectious (atherosclerosis, oncological diseases, neurodegenerative diseases such as Alzheimer’s disease, autoimmune diseases such as systemic lupus erythematosus, arthritis, etc.) diseases, as well as experimental surgery [26]. The data in the literature on the emergence of natural COVID-19 infection in domestic rabbits [27], as well as the successful intranasal infection of New Zealand white rabbits with SARS-CoV-2 [28], suggest that rabbits can be used as a model for the study of coronavirus infection. Since ACE2 is the major receptor through which the coronavirus penetrates host cells, blocking or disrupting the interaction between ACE2 and S protein may result in the reduced entry of viral particles [29].

## 2. Results and Interpretation

The comparison of the electrical properties of rabbit (rACE2) and human (hACE2) angiotensin-converting enzymes requires first the reconstruction of the three-dimensional (3D) structure of rACE2 and then the calculation of the surface electrical potential, which determines, along with the surface relief, the non-covalent binding energy of the peptides, as well as the association energy with the coronavirus S-protein.

### 2.1. Structural Analysis

As a first step, we compared the amino acid sequences of the polypeptide chains of the two ACE2 proteins corresponding to their genetic code. The chain is divided into three segments demarcated by vertical lines in Figure 2 (amino acid residues are numbered from the N-terminus): (a) signal peptide 1–18, (b) catalytic domain 19–615, and (c) transmembrane segment 616–805. The distinctive residues between hACE2 and rACE2 are colored based on their polarity and charge, with negatively charged hydrophilic aspartic acid (D) and glutamic acid (E) in red; positively charged lysine (K), arginine (R), and histidine (H) in blue; and uncharged hydrophilic serine (S), asparagine (N), threonine (T), glutamine (Q), and cysteine (C) in green; and the uncharged hydrophobic tryptophan (W), leucine (L), methionine (M), alanine (A), isoleucine (I), glycine (G), valine (V), phenylalanine (F), tyrosine (Y), and proline (P) in yellow. The identical amino acids are uncolored. Comparison of the primary amino acid sequence of the two proteins shows a high identity of 85% between the two chains (686 of 805 residues are identical, and 119 are different), and no deletions are present. As it can be seen on Figure 2, a great number of point mutations (41 of 197 amino acid residues are different) is concentrated in the transmembrane collectrin-like segment, but only 10 of them (less than one fourth) are associated with a change in hydrophobicity and hydrophilicity. The catalytic segment of both chains is composed of 597 amino acid residues, 520 of which are identical; the different 77 amino acid residues are 12.9%. The full identity of the amino acid residues in the enzyme active center (histidine 374, 378, 505; glutamic acid 375, 402, 406; and tyrosine 515) suggests a complete overlap in the functional aspects. In other words, rabbit ACE2 should work with the same substrates as human ACE2. However, the high compositional identity does not mean that the two chains are folded identically, i.e., it cannot be said in advance that their 3D structures are analogous.

To create a 3D model of the rabbit ACE2 protein, we applied computer mutational analysis, in which the different amino acid residues are sequentially substituted in the catalytic domain of human ACE2, and after each substitution, the new 3D coordinates of all atoms in the protein globule are calculated. The 3D models of the catalytic domains of the X-ray crystallographic hACE2 (PDB: 1r42) and the computer reconstructed rACE2 are shown in Figure 3 (the upper two models), colored in red (human) and green (rabbit). To compare the 3D structure of the rabbit (reconstructed) and human (taken from PDB) catalytic domains of ACE2, we used a computer program to align the 3D atomic coordinates of the two protein globules. The band model of ACE2 (Figure 3, the bottom left model) shows that the 3D structure of the two proteins is almost identical, there are only two segments where the tertiary structure differs, but this only applies to the unstructured segments of the chain, whereas the α-helical segments are identical. The bottom right model in Figure 3 shows the surface of the two aligned globules. Each atom is partially colored in red and green (the ratio of the two colors depends on the viewing angle), but the contours of the majority match, indicating that the coordinates of the unsubstituted atoms are virtually identical. The atoms of the different (substituted in rACE2) amino acid residues are colored entirely in red or green. Thus, the alignment (Figure 3, bottom two models) shows that the structure of the rabbit ACE2 is nearly identical to the human one both in the volume of the protein globule and on its surface.

### 2.2. Electric Charge

The electrical properties of the proteins are determined by the ability of 7 of the 20 amino acid residues to associate or dissociate a proton H^+^ in aqueous medium, whereby they acquire a positive or negative coulombic electrical charge, respectively (the partial charges due to polarization of covalent bonds do not contribute significantly, being dipoles whose electric field acts over a very small distance). The ionizable groups have different affinity for the protons, so they join or detach a proton at different concentrations of H^+^ in the medium. This dependence on the proton concentration in the medium results in a pH-dependent *nz* (pH) of the net charge *nz* (the difference between the positive and negative coulombic charges). At a strongly acidic pH, the carboxyl groups are in an undissociated form (–COOH) and the charge is determined only by the proton-associating N-containing groups: mainly the amino group (–NH_3_^+^) of lysine, the imidazole group of histidine, and the guanidine group of arginine. At a strongly alkaline pH, only the dissociated carboxyl groups (–COO^−^) have a charge, since the amino groups (–NH_2_) have lost the associated proton. At a neutral pH, the net charge is determined by the difference between the positively and negatively charged groups. Only the ionizable groups located on the surface of the protein globule have the ability to associate or dissociate a proton because only they are in contact with the aqueous medium where the proton is associated with a water molecule, forming a hydroxonium ion H_3_O^+^ which cannot penetrate into the interior of the globule. The local pH near the globule surface is different from the pH within the solution bulk because the surface electrostatic field with a negative or positive sign attracts or repulses protons, respectively. Therefore, each of the ionizable groups has an individual pK_a_, i.e., 50% of the time, it is charged or uncharged at a pH value that is different from the average for these kinds of groups. This means that the net charge of the protein globule is different from that of the polypeptide chain in its unfolded form (random coil conformation), but this difference is manifested in the pH regions where the association or dissociation of protons takes place: a moderate low pH for acidic groups and high pH for basic ones.

Taking the above considerations into account, we calculated the pH dependence of the net charge *nz* in the 3D conformation of the catalytic domain, using the crystallographic structure of hACE2 and the reconstructed rACE2, and accounting for the two positive charges of the Zn^2+^ atom in the active center of the enzyme (Figure 3). The pH-dependent *nz* (pH) have two steep sections where the net charge changes, being positive at an acidic pH and negative at an alkaline pH (Figure 4).

The catalytic domain of human ACE2 has 70 positive charges at pH 1 and 98 negative at pH 14, while the rabbit domain has 71 positive and 102 negative charges, respectively. I.e., rACE2 has one positive charge and four negative charges more than hACE2 at the two extremal pH values. However, the difference in the charge number does not shift the isoelectric point, whose value is pI 5.21 for the catalytic domain of both human and rabbit ACE2 in 3D conformation (the insert in Figure 4). At a physiological pH, the number of charges is significantly less because many of the chargeable groups, including the phenol hydroxyl group of tyrosine residues, remain neutral. At pH 7.4, the net charge of the two catalytic domains in 3D conformation is almost equal: 24.6 and 24.7 negative charges (the fractional numbers reflect the different degree of protonation over time). The above numbers of the coulombic charges are actual when the chain retains its 3D conformation (folded state), but at the extremal pH values, the protein is denaturated. In unfolded (random coil) conformation, the number of the charges is bigger because then every chargeable group, depending on its pK_a_ value and pH of the aqueous medium, can bear coulombic charge: positive for the proton-binding groups or negative for the proton-dissociating groups (Section 3.3).

### 2.3. Electrostatic Potential

The electric charges of the protein globule create an electrostatic field that is non- uniform due to the irregular distribution of the charges. The sign and value of the local electrostatic potential on the surface of the globule determine the interaction of ACE2 with peptides and other proteins. In Figure 5, one can see the electrostatic potential being unequal in sign and value on the part of the ACE2 surface where the enzyme active center is located. At pH 5.0, this surface is predominantly positively charged, and at pH 7.4, it is almost entirely negatively charged, except for the active center (the blue atoms in the center). This difference should influence the enzyme activity (Section 3.6).

### 2.4. Thermodynamic Stability

The change in the composition and sequence of the polypeptide chain affects the thermodynamic stability of the 3D structure of each globular protein. Any replacement of one amino acid by another can be considered as a point mutation that alters the stability of the globule because of the different hydrophilicity/hydrophobicity and the electric charge of the substituted and substituting amino acid residues [21]. Figure 6 shows the hydrophilic and hydrophobic residues of the entire polypeptide chain of the two enzymes hACE2 and rACE2, including the signal peptide, catalytic domain, and transmembrane segment, which sequentially fold during intracellular synthesis of the polypeptide chain. It can be seen that the number of substitutions of hydrophilic to hydrophobic amino acid residues and vice versa is not large, which means that the difference in primary structure (composition and sequence) cannot dramatically change the thermodynamic stability of the 3D structure of the two proteins.

A quantitative measure of the thermodynamic stability of the 3D structure of the protein globule is the folding energy Δ*G*_fold_, which is determined by the Gibbs free energy alteration at the transition from the fully unfolded state (random coil) to the native 3D structure of the protein globule at a given temperature, pH, and solution composition. Negative values of Δ*G*_fold_ mean that the free energy decreases at the transition; i.e., the 3D structure is thermodynamically advantageous. The folding energy Δ*G*_fold_ is determined by the contribution of the four types of forces with which the amino acid residues interact with each other and with the water molecules of the medium. The components of Δ*G*_fold_ determined by the energies of the hydration, hydrogen bonds, and Van der Waals forces are independent of the pH of the medium, but the electrostatic Δ*G*_el_ component is pH- dependent, since it is determined by the degree of H^+^ association or dissociation of the ionizable groups. Therefore, the pH dependence of *G*_el_ indicates the pH-range of maximal stability of the 3D structure.

Since the number and location of the ionizable groups are determined by the composition and sequence of the amino acid residues in the polypeptide chain, the difference in the primary structure of the human and rabbit ACE2 (Figure 2) means that their thermodynamic 3D stability should be different. Indeed, the pH-dependent Δ*G*_el_(pH) of the electrostatic component Δ*G*_el_ of the folding energy of the catalytic domain of hACE2 and rACE2 are quite different by the absolute values but have almost analogical course (Figure 7). Considering that the isoelectric points pI 5.2 coincide and the net charge is almost equal in the range pH 6–9 (Section 2.2, Figure 4), it can be inferred that the difference in the Δ*G*_el_ values reflects the different 3D location of the coulomb charges in the two protein globules. The difference in the absolute values of Δ*G*_el_ (Figure 7) demonstrates the sensitivity of the electrostatic component of the folding energy to the 3D structure of the protein globules (Section 3.2). The negative values of Δ*G*_el_ mean that the electrostatic energy stabilizes the 3D structure of the hACE2 domain in the pH range 5.5–9.8. In the case of rACE2, this range is slightly wider: pH 5.1–10.0. Both enzymes are most stable at pH 6.5 (the deeper minimum), where the folding energy is maximal by absolute value.

The positive values of Δ*G*_el_ at pH < 5.5 and pH > 9.8 for hACE2 and also at pH < 5.1 and pH > 10.0 for rACE2 mean that in these extreme pH ranges, the 3D stability of the protein globule is reduced, since electrostatic repulsion between predominantly positive (at acidic pH) or negative charges (at alkaline pH) dominates there. However, the positive values of Δ*G*_el_ does not imply that the polypeptide chain necessarily transforms to the unfolded state (random coil) because the main contribution (about 90%) in the folding energy Δ*G*_fold_ gives its hydration component, due to which the aqueous medium pushes the hydrophobic amino acid residues into the core of the protein globule, while the hydrophilic ones remain on its surface. The reduction of Δ*G*_fold_ (at positive values of the electrostatic component Δ*G*_el_) is rather weak due to the relatively small contribution of the electrostatic interactions in 3D conformation of the polypeptide chain. I.e., the predominant electrostatic repulsion (in the two extremal pH ranges where Δ*G*_el_ has positive values) reduces the 3D structural stability of the protein globule and it can be denaturated in some degree, up to the state of melt globule, when the polypeptide chain retains its globular conformation but loses the secondary structure (α-helix and β-sheets).

## 3. Discussion

### 3.1. Collectrin Similarity

In the literature, the transmembrane domain of ACE2 is described as a collectrin-like [30]. To verify this claim, Figure 8 (top rows) shows a part of the hACE2 chain (including the transmembrane segment) as compared to the human collectrin chain (bottom rows). Taking into account the presence of deletions, the alignment was made to maximize the matching of amino acid residues in the two chains. Comparison of this part of the human ACE2 chain shows 82% similarity of the amino acid residues composition, but only 40% identity of their sequence with those of collectrin, which is not sufficient to claim the structural similarity of the two proteins. Because the hydrophobicity of the amino acid residues is particularly important, their cells in Figure 8 are colored yellow, in contrast to those of the hydrophilic residues (green colored). This distinction allows identification of the hydrophobic segment of the two chains (the long, yellow-stained segment). The comparison shows that this part of the hACE2 chain is almost identical in composition and amino acid sequence to the human collectrin [31,32]. This similarity suggests that the structure of the intramembrane segment of the two proteins is also similar, although for now its 3D conformation remains unknown. The other two parts have alternating short segments of mixed hydrophilicity, suggesting that they are extramembrane: the upper segment continues to the catalytic domain of hACE2 (extracellularly located), and the lower one is a small intracellular hydrophilic segment. Collectrin (also called Tmem27) is a transporter protein which plays an important role in neutral amino acid transport in the kidneys, interacting with proteins of the Slc6 family (in particular B^0^AT1, B^0^AT3) [33]. Like collectrin, ACE2 also participates in amino acid transport but in the small intestines, where it interacts with B^0^AT1 [34,35]. Considering the similarity not only in primary amino acid sequence, but also in functional aspect, the claim that the transmembrane segment is collectrin-like can be considered appropriate.

### 3.2. Stability of 3D Structure

The stability of a three-dimensional (3D) protein globule structure is determined by four types of forces: hydrogen bonds, hydrophilic, electrostatic, and van der Waals. Intramolecular hydrogen bonds determine the secondary structure: α-helix and β-sheet. The hydrophilicity of the amino acid residues is determined by their ability to form hydrogen bonds with water molecules. The Van der Waals forces are attractive, their main component being the London dispersion forces, while the partial charge interactions (dipole–dipole and dipole-induced dipole) play a minor role. The electrostatic forces are both attractive between the coulombic charges of the opposite sign (positive to negative) and repulsive when the charges have the same sign (positive or negative). The spatial folding of the polypeptide chain is determined primarily by the affinity of the amino acid residues for water molecules: hydrophilic residues remain on the surface of the protein globule, whereas hydrophobic residues are predominantly in the globule core. This distribution is thermodynamically determined: the free energy decreases when the polypeptide chain folds to form a globule with a hydrophobic core and a hydrophilic shell. This ensures a maximum number of contacts of water molecules from the medium with the electronegative atoms (O and N) with which they form hydrogen bonds (HO–H…O– and HO–H…N–), and no contact in the hydrophobic residues with which hydrogen bond formation is impossible due to the absence of such atoms. The degree of hydrophilicity varies depending on the ratio of hydrophilic to hydrophobic (–CH_2_–) groups in the amino acid residues, which determines the sign and value of the free energy change upon transfer of an amino acid residue from an organic to aqueous solution [36].

In the Protein Data Bank (PDB), for each deposed 3D structure, usually there are several models obtained with different methods: X-ray crystallography of the protein crystal, nuclear magnetic resonance (NMR), and cryo-electron microscopy (cryo-EM). Crystallographic models are the most reliable and have the highest resolution, but they have the disadvantage that in the crystal, the protein globules are tightly packed, and therefore unstructured segments of the polypeptide chain may have a different spatial arrangement in aqueous media. We choose the 3D protein models using as a criterion the pH-dependent Δ*G*_el_(pH) of the electrostatic component Δ*G*_el_ of the folding energy, which indicates how thermodynamically advantageous the 3D structure of the protein globule is in dilute aqueous solution when the influence of neighboring globules is negligible. These dependences are very sensitive to alterations in the 3D structure of the protein globule because Δ*G*_el_ is the energy of a system of coulombic charges: the shift of every element causes alterations to its value.

Figure 9 shows the Δ*G*_el_(pH) dependencies of the catalytic domain of rabbit ACE2: the reconstructed (on the base of the original 3D model PDB:1r42 of hACE2) and two optimized rACE2 models. The first model has entirely positive values of Δ*G*_el_ (curve 1), suggesting that it is not the most thermodynamically advantageous 3D structure. To approximate the 3D structure of rACE2 to the real one in aqueous media, we optimized the model, whereby the unstructured segments receive atomic coordinates corresponding to a thermodynamically more favorable conformation with minimal free energy. The optimization performed with two computer programs discloses that better results (a deeper minimum of the pH-dependent Δ*G*_el_(pH) of the electrostatic component Δ*G*_el_ of the folding energy) gives the YASARA program, in comparison with Chimera. The pH dependence of the optimized model (curve 3 in Figure 9) shows that the Δ*G*_el_ values become negative in the range pH 5.1–10.0 with minimum at pH 6.5. Consequently, the 3D structure of the catalytic domain of rACE2 is the most thermodynamically stable in this pH range.

The reconstruction of the rACE2 catalytic domain, based on hACE2, is analogous to that used for single point mutations in viral proteins [21]. Since the 3D structure of the catalytic domain of hACE2 is obtained using a recombinant protein synthesized considering the genetic code of only part of the polypeptide chain (indicated by vertical lines in Figure 2), the model of the catalytic domain probably does not include the entire off-membrane hydrophilic part of the enzyme. The models, shown in Figure 3, are colored in red and green according to the human and rabbit specific diet: red meat and green grass. The almost complete coincidence of the 3D structure of the catalytic domains of the human and rabbit ACE2 can be explained by the fact that the substituted amino acids have commensurate side group volumes, which gives them sufficient freedom to form an α-helical secondary structure. The exception is proline, whose conformational angles are highly constrained and there the helical segments are broken

Figure 9 presents the electrostatic components Δ*G*_el_ of the folding energy calculated for 3D models of rACE2, which are reconstructed by two different approaches. The first one is based on an experimental 3D model of hACE2, which is modified by mutagenesis according to the primary amino acid sequence of rACE2 (curve 1), after which the model has been optimized to reach the most energetically advantageous atomic coordinates (curves 2 and 3). The second approach is based on the construction of a 3D model beginning directly from the primary amino acid sequence of rACE2, for which purpose we employed the program AlphaFold2. The practical coincidence of curves 3 and 4 shows that 3D models of rACE2, obtained by these two alternative computer techniques, have practically equal thermodynamic stability. The coincidence reflects almost the same coordinates of the coulomb electric charges of the groups ionized at a given pH. The alignment of the two 3D models of rACE2, which compares the atomic coordinates of two protein globules, shows that the root mean square deviation (RMSD) is 0.199 nm, and the template modeling score (TM-score) is 0.95. These values mean that the two rACE2 models practically coincide (values in the ranges RMSD ≤ 0.2 nm and TM-score 0.5–1.0 are designed as a good coincidence).

### 3.3. Electric Charge in Unfolded and 3D-Folded Conformation

The coulombic electric charges of the catalytic domain in the unfolded conformation of the polyelectrolyte chain (random coil, without Zn^2+^) are 69 and 70 positive at pH 1, or 111 and 115 negative at pH 14 for hACE2 and rACE2, respectively. These numbers correspond to all ionizable (chargeable) groups which can obtain coulombic charges as a result of protonation or deprotonation at a given pH. The amino acid residues of human ACE2 catalytic domain, which can bear positive charges, are 69, including 34 lysine, 18 arginine, 16 histidine, and N-end amino group; the negatively chargeable residues are 111 (33 aspartate, 47 glutamate, 28 tyrosine, two cystine (six cystine form disulfide bridges), and C-end carboxyl group). The chain of the rabbit ACE2 catalytic domain has 70 positively chargeable residues (33 lysine, 23 arginine, 13 histidine, and N-end amino group) and 115 negatively chargeable: 31 aspartate, 53 glutamate, 28 tyrosine, two cystine (six cystine form disulfide bridges), and C-end carboxyl group.

In 3D conformation, the positive charges are 68 and 69 (at pH 1), and the negative charges are 100 and 104 (at pH 14), respectively, for the catalytic domains of hACE2 and rACE2 without Zn^2+^ (Section 2.2). So, at these extreme pH values, the numbers of coulombic charges of the folded chain are less with one positive and 10 negative charges, as compared to those in the unfolded conformation for both hACE2 and rACE2. This difference can be explained by the following. Firstly, the number of the chargeable groups is calculated at extreme pH values, but at a physiological pH the number of the charged groups (bearing coulombic electric charges) is significantly less because of a lower degree of ionization, which is determined mainly by the dissociation constant pK_a_ of a given ionizable group (the specific pK_a_ value can obtain extreme values). The groups with a high degree of charging at pH 7.4 are the carboxylic groups of the asparagine and glutamine acid residues (pK_a_ 3.0–4.7, negatively charged) and the positively charged amino group of lysine (pK_a_ 9.4–10.6) and the guanidine group of arginine residues (pK_a_ 11.6–12.6). The phenol group of tyrosine (pK_a_ 9.8–10.4) is deprotonated at pH 14, but protonated (uncharged) at neutral pH. In addition, only about half of the imidazole groups of the histidine residues (pK_a_ 5.6–7.0) are protonated; the rest are uncharged. Secondly, some of the dissociable groups catch a proton and become uncharged when the chain folds into a 3D globule. In the hydrophobic core of the protein globule, the dielectric permittivity ε ≈ 4 is very low, and consequently the electrostatic attraction between COO^−^ and H^+^ is about 20 times stronger than those in the aqueous medium with ε = 80. Therefore, the deeply located carboxylic groups usually are uncharged even at pH 14.

The disappearance of one positive charge (from 69 to 68 for hACE2, and from 70 to 69 for rACE2) at pH 1 when the polypeptide chain folds in the 3D globule is surprising because at such high proton concentration (pH 0 at 1 mol/L H_3_O^+^), all positively chargeable groups, both located on the surface and in the hydrophobic core (where they more strongly retain the caught proton due to the low dielectric permittivity), must be protonated. We supposed that this disappearance can be explained by the emergence of one negative charge because of deprotonation of a surface-located carboxylic group having an extremely low dissociation constant pK_a_ < 1, instead the usual pK_a_ 3.0–4.7. Such a low dissociation constant can appear as a result of a strong positive electrostatic potential around these groups in 3D conformation of the polypeptide chain. This leads to a very low concentration of H_3_O^+^ cations in the vicinity of this carboxylic group, and therefore it loses its proton and becomes negatively charged even at pH ≤ 1. We have found that the suspect is the carboxylic group of Glu-402 residue with calculated pK_a_ ≈ 0, positioned in the surface pocket of the enzyme active center, where it is surrounded by positively charged groups. Due to its negative charge, this carboxylic group forms a coordinative bond with the Zn^2+^ atom when the polypeptide chain is in 3D conformation.

### 3.4. Isoelectric Point in Unfolded and 3D-Folded Conformation

In unfolded (random coil) conformation, all chargeable groups of the polypeptide chain that are in contact with the aqueous medium can bear a positive or negative coulombic charge due to proton association or dissociation in the dependence on pH and the dissociation constant pK_a_ (pH at 50% protonation). The intrinsic dissociation constant pK_a_ is then determined by the average degree of protonation for all groups of a given type. When the chain is folded into a 3D globule each chargeable group obtains an individual (apparent) pK_a_ value which can be defined by the equality of the times when the group is protonated or deprotonated. The difference between the intrinsic and apparent pK_a_ constants is due to the different local proton concentrations caused by the surface electrostatic potential in the vicinity of the group, which repels or attracts the H_3_O^+^ cations dependent on its sign. I.e., the charged groups create an electrostatic potential which influences the degree of their protonation in an acidic pH range for carboxylic (H^+^-dissociable) or in an alkaline pH range for amino (H^+^-associable) groups. The isoelectric point pI (pH at zero net charge, equality of the positive and negative coulombic charges) of the polypeptide chain is different in its unfolded (random coil) and folded (3D globule) conformation because of two reasons: (a) some of the chargeable groups can be located not on the surface of the globule, but in its hydrophobic core; and (b) the degree of protonation depends on the local electrostatic potential. In the case of ACE2, a third reason appears: upon folding into the 3D structure, seven amino acid residues form an active center (Section 2.1), which bind coordinately a Zn^2+^ cation, and respectively, the net charge of the protein globule increases by two coulombic charges.

The different number of the chargeable amino acid residues of the catalytic domains of the human and rabbit ACE2 (Section 3.3) leads to different pI values in the unfolded conformation of their polypeptide chains: pI 5.03 and pI 4.97, respectively, for hACE2 and rACE2; the lower rabbit pI is due to the bigger number of negatively charged groups. At the folding of the chain into 3D conformation, the isoelectric points become closer: pI 5.12 and pI 5.13, and shift to higher pH values owing to the disappearance of 11 negative charges. When the Zn^2+^ cation is bound in the active center of the 3D globule, the isoelectric points coincide: pI 5.21 (Figure 10).

On the contrary, at the unfolding of the polypeptide chain of the catalytic domain from 3D conformation to random coil (full denaturation), the isoelectric points shift to lower pH values with 0.17 and 0.23 pH units for human and rabbit ACE2, respectively. This phenomenon is caused by the disappearance of the surface electrostatic potential of the former ACE2 globule; consequently, the concentration of H_3_O^+^ cations (local pH) becomes almost equal to that in the bulk of the protein solution and the apparent pK_a_, especially those with extreme values, shifts to the intrinsic pK_a_ values of the chargeable groups.

### 3.5. Isoelectric Point: Calculated and Experimental

In our previous study [37,38], the experimentally determined isoelectric mean point pI 4.0 (two bands with pI 3.9 and pI 4.1 in isoelectric focusing gel) of the recombinant human ACE2-Fc (with Fc-tag attached to the C-end of the hACE2 polypeptide chain) was found to be two pH units lower than the calculated pI 5.99 (taking into account the charges of the Fc-tag). There are two reasons for this. The first is that the experiment was performed by isoelectric focusing in a medium with 8 mol/L urea, in which a complete denaturation occurs: the irreversible transition from the 3D structure state to the fully unfolded conformation (random coil). The influence of the polypeptide chain conformation on the isoelectric point (Section 3.4) is relatively small and quite insufficient to explain the difference of two pH units between the calculated pI 6.0 and experimentally measured pI 4.0 values of the isoelectric point.

The second reason for the two pH unit discrepancy that we found between the calculated and measured isoelectric points is that in the isoelectric focusing experiment, we have used recombinant hACE2 expressed in embryonic human cell culture with a high glucose concentration in the medium. In these conditions, an enzymatic glycosylation and non-enzymatic glycation occur (Section 3.8). Both processes shift the isoelectric point pI of proteins to lower pH values due to a reduction of the positively charged amino groups and emergence of additional negatively charged carboxylic groups.

### 3.6. Electrostatic Potential and Enzyme Activity

The surface electrostatic potential is dependent on the pH of the surrounding aqueous medium because of protonation or deprotonation of the chargeable groups of the amino acid residues located on the surface of the protein globule. The pH-dependent potential of ACE2 (Figure 5) influences its enzyme activity by strengthen or weakening the electrostatic interactions with the positively charged angiotensin peptides, since the enzymatic reaction is preceded by electrostatic attraction of them to, and adsorption onto, the negatively charged surface of the protein globule, as well as of the negatively charged C-terminal carboxyl group to the positive charges in the active center, where it forms a complex with the positively charged Zn^2+^ atom. The angiotensin I, II, and 1–9 (Figure 1) carry four, three, and four positive charges, respectively. The following suppositions can be made.

Firstly, the enzyme is differentially active depending on the localization of the epithelial cells to whose cytoplasmic membrane it is integrated and oriented extracellularly. ACE2 is most active in blood vessels, where pH 7.4 determines the highly negatively charged surface of the enzyme (Figure 5), adsorbing electrostatically the positively charged peptide. In the lumen of the upper respiratory vessels ACE2 is weakly active or even inactive, since there the pH of the secretion is lowered to pH 5.5–6.5 [39,40,41,42] because the CO_2_ (dissolved from the air) forms the carbonic acid H_2_CO_3_ whose dissociation to HCO_3_^−^ reduces the pH.

Secondly, the ACE2 enzyme must be differentially active toward the three peptides (Figure 1). Considering only the surface electrostatic potential of the catalytic domain (Figure 5, pH 7.4), the enzyme should be most active with angiotensin-I due to its three positive coulombic charges in the chain. This peptide binds most strongly to the negatively charged (at pH 7.4) surface and to the positively charged Zn^2+^ atom in the active center by the negatively charged carboxylic group of the C-terminal amino acid residue. Angiotensin-II is less readily adsorbed because it has only two positively charged groups in the chain. Angiotensin-1–9 also has two positive charges in the chain, but its C-terminal amino acid residue is neutral because it carries both a positive and negative charge and therefore binds weakly to the Zn^2+^ atom in the active center.

### 3.7. Verification of the 3D Reconstruction Procedure

In the present work, we have developed a procedure for the reconstruction of a 3D model of a globular protein whose amino acid sequence is known, and for which there exists an analogical protein with a determined 3D structure (obtained by X-ray crystallographic, NMR, or cryoEM techniques), which we refer to as the “basic protein”. For this purpose, we compare the polypeptide chains of the two proteins, called, respectively, basic and mutant, and then substitute the different amino acid residue in the basic 3D model, assuming that they are point mutations [21]. The approach is applicable when the two polypeptide chains are similar and there are no deletions and insertions. Then, we use a computer program for mutational analysis which generates the atomic coordinates of the substituting amino acid residues in the basic 3D structure. In this process, the atomic coordinates of the surrounding amino acid residues can be slightly shifted. Repeating this procedure step-by-step, we substitute all the different (mutant) residues. The individual shifts are summed up, which can lead to the appearance of a methodological artifact when the number of mutant residues is large. Additionally, we optimize the 3D structure with another program which minimizes the free energy of the protein globule by shifting the atomic coordinates. This raises the question of how different the reconstructed 3D structure is from the real one.

To validate the 3D reconstruction procedure, we have to find a couple of two analogical proteins (a basic and a mutant) with an experimentally determined 3D structure, substitute the different (mutant) amino acid residues in the basic 3D model, and compare the reconstructed 3D model of the mutant protein with its experimentally determined 3D structure. Finding such a couple of proteins with a reliably determined 3D structure is not a trivial task, because different 3D models have been deposited in the Protein Data Bank for the majority of proteins and it is not clear which one is correct. To select a model, we calculate the electrostatic component Δ*G*_el_ of the folding energy Δ*G*_fold_, using it as a criterion for correctness of the models. This criterion is based on the conception that Δ*G*_el_ is determined by the free energy of a 3D system of coulombic charges; the shift of their coordinates alters the Δ*G*_el_ value. The absolute value of Δ*G*_el_ and its pH-dependent Δ*G*_el_(pH) are very sensitive to small changes in the 3D structure of the protein globule, as these emerge by the optimization of the 3D structure (Figure 9). Then, the model with maximal 3D stability is the one that has the minimal Δ*G*_el_ (maximal in absolute value with negative sign, a deeper minimum of the Δ*G*_el_(pH) dependence), which is determined by the most energetically optimal location of the coulomb charges of the protein globule.

Assuming that the crystallographic models of well-known proteins are the most reliable, we have chosen the horse myoglobin which has a well-known human analogue. For these two human–animal proteins, we have reconstructed the 3D model of the horse analog by the substitution of 18 mutant residues in the human 3D model, according to the difference in the amino acid sequences of the two proteins. The root mean square deviation (RMSD) of the atomic positions of the backbone atoms of original and reconstructed horse myoglobin is 0.076 nm (values under 0.1 nm indicates very high degree of identity). The coincidence of the isoelectric points pI 6.7 and the pH-dependent *nz*(pH) of the net charge *nz* of the original and reconstructed 3D models of the horse myoglobin (Figure 11, Insert) are evidence that all mutations are included correctly. Figure 11 shows very close courses of the pH-dependent Δ*G*_el_(pH) of the electrostatic component Δ*G*_el_ of the folding energy of the original and the reconstructed 3D models of horse myoglobine. Taking into consideration the high sensitivity of Δ*G*_el_(pH) dependence to a small shift in the atomic coordinates, and also that the optimization procedure is not performed, the nearness of the two pH curves shows that the 3D models (original and reconstructed) can be assumed as practically identical. This verification confirms the procedure for the reconstruction of the 3D model of the rabbit (mutant) protein by substitution of the different amino acid residues in the 3D model of the human protein (Section 2.1).

### 3.8. Glycosylation and Glycation

Glycosylation is a post-translational modification of the proteins which takes place in the Golgi apparatus. This process is enzymatically mediated by glycosyltransferases and results in the formation of N- or O-glycans [43]. The tree-branched sugar structures formed by glycosylation terminate in mammals with one to four sialic acid residues whose carboxyl groups are deprotonated (COOH → COO^−^) at physiological pH values, and this results in the appearance of up to four negative charges for each modified amino acid residue which shifts the isoelectric point to a low pH [44,45,46,47,48]. Conversely, the enzymatic removal of sugar residues results in a decrease in molecular mass and an increase in isoelectric point [49,50,51,52].

In contrast to glycosylation, glycation is a non-enzymatic, spontaneous reaction, in which the free amino groups (–NH_2_) of lysine and asparagine amino acid residues interact with the carbonyl group of monosaccharides (such as glucose, fructose, etc.) [53,54]. This leads to the formation of a Schiff base which undergoes rearrangement and finally an Amadori product is obtained. The consequence of the glycation is that as the number of the charged amino groups (–NH_3_^+^) in the protein decreases, the positive coulombic charges also respectively reduce, and the isoelectric point shifts to a low pH.

In our previous experiment with isoelectric focusing (Section 3.5), we have used recombinant hACE2 expressed in embryonic human cell culture (line HEK 293). There is evidence in the literature indicating the presence of various sugars (fucose, mannose, galactose, N-acetylglucosamine) bound to proteins synthesized by this recombinant technology [55]. Because of the high glucose concentration in the medium of the cell culture, both processes of glycosylation and glycation are reinforced, leading to an increase in the molecular mass and a change in the electric charge. The used recombinant hACE2 catalytic domain contains 723 amino acid residues (from 18 to 740, numbered from N-end, Figure 2) and covalently connected Fc-tag, totaling 952 residues. The molecular mass (calculated according to the amino acid content) was 83.68 kDa and 109,4 kDa with and without Fc-tag, respectively. The molecular mass of this recombinant protein (hACE2+Fc-tag), experimentally measured by gel electrophoreses, is 145-150 kDa, according to the data of the producer. The increment 36–41 kDa is caused by the attached sugars.

## 4. Methods

The analysis of the analogy in amino acid content and sequence of the polypeptide chain of human (hACE2) and rabbit (rACE2) angiotensin-converting enzyme-2 was performed using their published primary sequences obtained by analysis of the DNA sequence of the corresponding gene [56]. The crystallographic 3D model of the catalytic domain of hACE2 with code 1r42 in the Protein Data Bank (PDB) was used for the reconstruction of its rabbit analog. The 1r42 model is a recombinant protein expressed in the system based on the butterfly *Spodoptera frugiperda* and crystallized in aqueous solution with high ionic strength 100 mM Tris-HCl (pH 8.5), 200 mM MgCl_2_, and 14% polyethylene glycol 8000 [57]. The atomic coordinates of this model were optimized by the computer programs Chimera [58] and YASARA [59] to find the energetically most advantageous conformation of the flexible side-chain amino acid residues under physiological conditions in an aqueous medium: pH 7.4 and ionic strength 0.15 mol/L NaCl, at which the free energy of the protein globule is minimal. The 3D atomic coordinates of rabbit ACE2 were obtained by the replacement of the different amino acid residues using the program for in silico mutagenesis DUET, combining two programs: the Site Directed Mutator (SDM) and a program for missense mutations, called mCSM [60,61,62], which finds the 3D atomic coordinates in the protein globule after the replacement of an amino acid residue in hACE2 structure with the corresponding residue of rACE2 in its polypeptide chain. The mutation procedure was repeated step-by-step to replace all different residues. By this, three 3D models of rabbit ACE2 were obtained: one based on the original hACE2 (PDB: 1r42), and two optimized. To select the most appropriate one for further calculations of the electric and thermodynamic properties, the six models (three human and three rabbit) of the catalytic domain of ACE2 were examined by calculation of the electrostatic component Δ*G*_el_ of their folding energy Δ*G*_fold_, using as a criterion the maximal absolute value of the decrease in the Gibbs free energy at the transition from fully unfolded (random coil) to the 3D structure (protein globule) of the corresponding model. The results of this examination disclosed that the most energetically advantageous (with minimal free energy) are the two models obtained by optimization using the program YASARA. The results shown in this article, including 3D models, the net charge, the surface electrostatic potential, and the electrostatic folding energy Δ*G*_el_, correspond to these two models. The program AlphaFold2 [63] was used for the independent construction of a 3D model of rACE2 using only its primary amino acid sequence.

The computer programs for macromolecular alignment SuperPose [64] and TM-score [65] were used to compare the 3D structure of the catalytic domains of the hACE2 and the reconstructed rACE2. The computer program for protein electrostatics Bluues [66] was employed to calculate the pH-dependent electrostatic component Δ*G*_el_ of the free energy of folding. The program Propka [67,68] for protein electrostatics was employed to compute the electric parameters of the catalytic domains of hACE2 and rACE2: pH dependence of the net charge, the isoelectric point (zero net charge), and the local surface electrostatic potential at a given pH. The programs Chimera [58], PBEQ Solver [69,70], and VMD: Visual molecular dynamics 1.9.2 [71] were used to visualize the 3D models and the electrostatic potential on the surface of the protein globules.

## 5. Conclusions

Comparison of the primary amino acid sequence of the full polypeptide chains (including the signal peptide, catalytic domain, and the transmembrane segment) of human ACE2 (hACE2) and rabbit ACE2 (rACE2) shows that 85% of the residues are identical (686 of 805 amino acids). A high degree of similarity is also observed by 3D macromolecule alignment at the tertiary structure: the α-helical segments are identical; only two unstructured segments have noticeably different coordinates. The pH dependences of the electrostatic component Δ*G*_el_ of the folding energy of the two catalytic domains disclose that the 3D structure of the two proteins is maximally stable at pH 6.5. The negative values of Δ*G*_el_ indicate that the intramolecular electrostatic interactions contribute to the 3D structural stability (determined mainly by the hydration energy) in the range pH 5.5–9.8 for hACE2 and 5.1–9.8 for rACE2.

The number of positively chargeable groups of the unfolded polypeptide chains of the catalytic domains of hACE2 and rACE2 are 69 and 70, and the negatively chargeable ones are 111 and 115, as calculated at pH 1 and pH 14, respectively. The charge number depends on the conformation of the polypeptide chain: the 3D catalytic globules have 68 and 69 (pH 1) positive charges, and 100 and 104 (pH 14) negative charges, respectively, for hACE2 and rACE2. I.e., both catalytic domains have one positive and 10 negative charges less in 3D-folded conformation, as compared to the fully unfolded chain (random coil). The difference is due to the different degree of ionization of the same type of chargeable groups on the surface (in contact with aqueous medium) and being submerged in the hydrophobic core of the protein globule. In contrast to the different number of the chargeable groups, in 3D conformation the isoelectric points pI 5.21 of hACE2 and rACE2 coincide, and their net charges are almost equal: about 25 negative coulombic charges at pH 7.4. The electrostatic potential is likewise distributed on the surface of the two catalytic globules, but it is unhomogeneous and strongly dependent on the pH of the aqueous medium: at a low pH, the potential is almost positive, but strongly negative at pH 7.4. Considering the positive charges of the angiotensin peptides, the pH-determined difference of the surface potential suggests that the ACE2 enzyme is most active at pH 7.4.

The significant similarity in 3D structures, thermodynamic, and electric properties of the catalytic domains of hACE2 and rACE2 suggests that rabbits can be successfully used as an animal model for the investigation of both infectious and non-infectious diseases with involved ACE2, including the coronavirus infection, allowing the results to be extrapolated to humans, and also for the in vivo investigation of various agents that disrupt the interaction of ACE2 with the coronavirus S-protein, such as potential antiviral drugs.

## Figures and Tables

**Figure 1 ijms-25-12393-f001:**
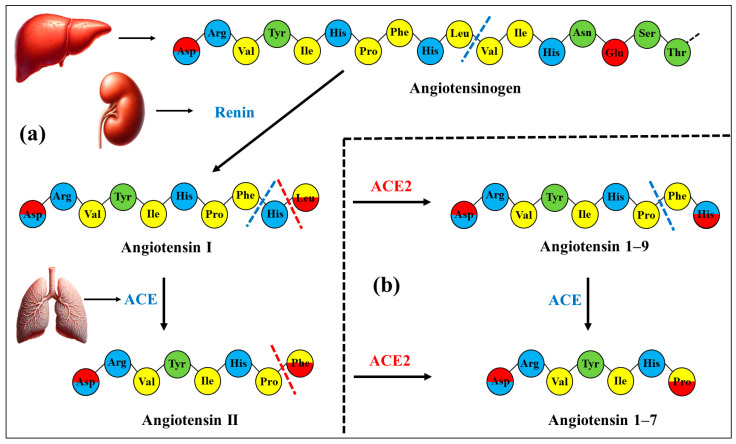
Renin-angiotensin system. *Subsection* (**a**) on the left: Renin and angiotensin-converting enzyme (ACE) action. The precursor α-2-globulin angiotensinogen is produced by hepatocytes. The renal enzyme renin cleaves the covalent peptide bond after the first 10 amino acids from the N-terminus of angiotensinogen, leading to the formation of angiotensin-I (first reaction). This decapeptide is converted by the pulmonary angiotensin-converting enzyme (ACE) to the octapeptide angiotensin-II by the cleavage of the last two amino acid residues, resulting in the emergence of a carboxylic group on the C-terminus (second reaction). *Subsection* (**b**) on the right: Angiotensin-converting enzyme-2 (ACE2) action. The angiotensin-converting enzyme-2 (ACE2) cleaves one amino acid residue from the C-terminus of both angiotensin peptides, which leads to the formation of the nonapeptide angiotensin-1–9 and the heptapeptide angiotensin-1–7, respectively. The cleaved peptide bonds (in the two subsections) are shown by dotted lines colored according to the corresponding enzyme: blue (ACE) or red (ACE2). The amino acid residues are colored according to their charge and hydrophilicity: green (uncharged hydrophilic), blue (positively charged hydrophilic), red (negatively charged hydrophilic), and yellow (uncharged hydrophobic); the charges are determined at neutral pH. The end-side residues are marked by double color considering the protonated α-amino group (NH_3_^+^–) on the N-terminus and the deprotonated carboxylic group (–COO^−^) on the C-terminus. The N-end aspartic acid (Asp) residue bears one positive (NH_3_^+^–) and one negative (COO^−^–) charge. The C-end histidine (His) residue of the angiotensin-1-9 bears one negative charge (the deprotonated carboxyl group) and one positive charge at acid pH (the protonated imidazole group), which disappears at basic pH. The C-ends of the remaining three peptides have only negative charge (–COO^−^).

**Figure 2 ijms-25-12393-f002:**
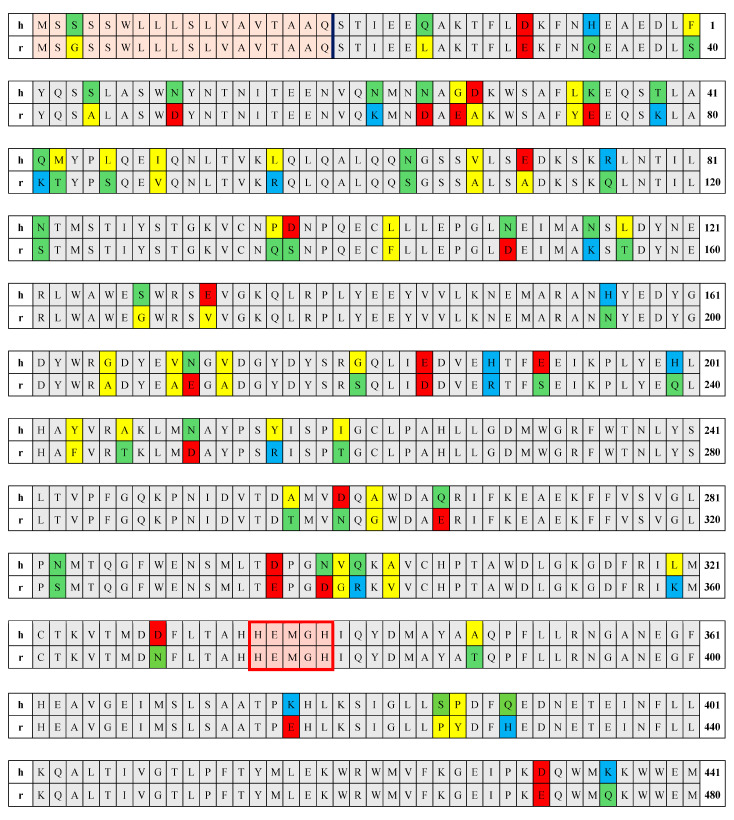
Primary amino acid sequence of the polypeptide chains of human (the rows beginning with **h**) and rabbit (**r**) ACE2. The numbers at the end of every row indicate the first and the last amino acid residue on the corresponding row. The amino acid residues are denoted according to the standard one-letter code. The cells of the different residues are colored according to the electric charge at pH 7.0 and the hydrophilicity of the given residue: red (negatively charged hydrophilic), blue (positively charged hydrophilic), green (uncharged hydrophilic), and yellow (uncharged hydrophobic). The polypeptide chain is divided (indicated by vertical lines) into three segments: signal peptide (amino acid residues 1–18, colored in bright orange), catalytic domain (19–615, colored in bright gray), and transmembrane segment (616–805 colored in bright purple). The five residues included in the zinc-binding motif HEMGH of the active center of the enzyme are denoted by the red rectangle.

**Figure 3 ijms-25-12393-f003:**
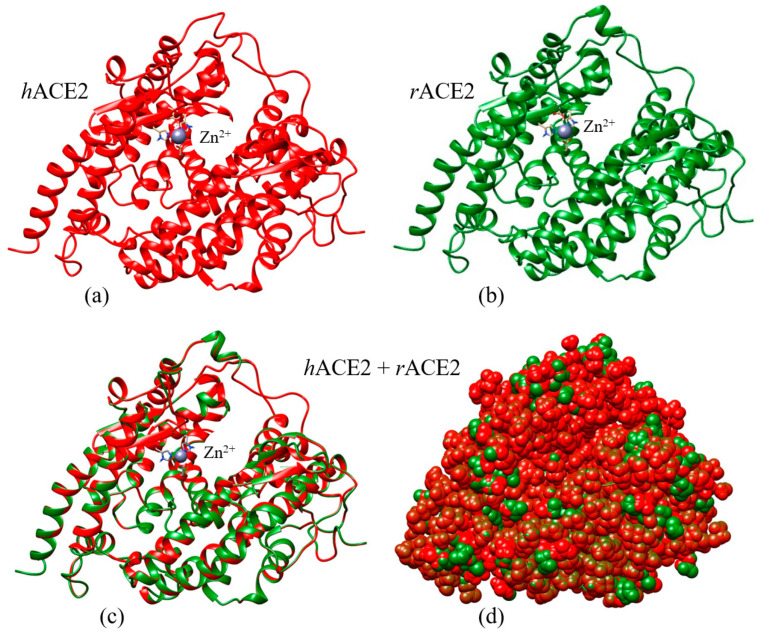
(**a**–**d**) 3D structure models of the human (red, hACE2) and rabbit (green, rACE2) catalytic domains of the angiotensin-converting enzyme-2 (the upper two models (**a**,**b**)); the α-helix segments are depicted as ribbon spirals. The model of the rabbit ACE2 (**b**) is reconstructed by replacement of the different amino acid residues in the hACE2 (PDB: 1r42) model (**a**). The violet spherical object (Zn^2+^) is the zinc atom in the enzyme active center. The low two models (**c**,**d**) present the aliment (hACE2+rACE2) of the human and rabbit ACE2 in two different projections: the 3D volume of the protein globules (the ribbon model on the left) (**c**) and the 2D surface of the globules (the atomic model on the right) (**d**). The right bottom model (**d**) presents the atoms exposed on the surface of the aligned two protein globules; the atoms are colored fully in red (hACE2) or green (rACE2) when they are entirely protruded above the others, or in mixed color when their coordinates partially coincide. The brightness, shade, and color nuance of the atomic images give the impression for a quasi 3D surface of the protein globules.

**Figure 4 ijms-25-12393-f004:**
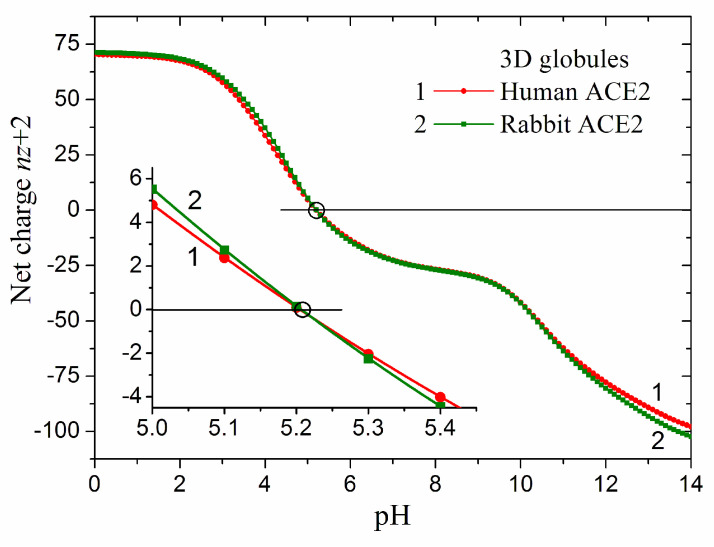
pH dependences of the net electric charge *nz* + 2 of the globular catalytic domain of human (PDB: 1r42, hACE2, red curve 1) and reconstructed rabbit (rACE2, green curve 2) angiotensin-converting enzyme-2 in 3D conformation of the polypeptide chain in aqueous medium. The net charge of the two globular domains is the algebraic sum of the positive and negative coulombic charges of the polypeptide chain with the addition of two positive charges of the Zn^2+^ atom in the catalytic center. *Insert*: pH dependence of hACE2 (red curve 1) and rACE2 (green curve 2) with denoted isoelectric point (*nz* = 0): pI 5.21 (human) and pI 5.21 (rabbit) ACE2.

**Figure 5 ijms-25-12393-f005:**
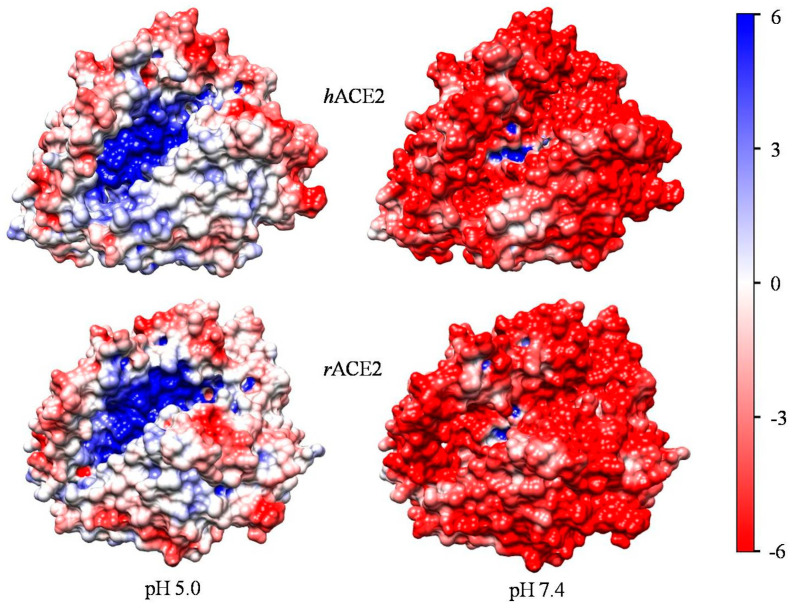
Electrostatic potential on the 3D surface of the catalytic domain of human (hACE2, two upper models, PDB: 1r42) and reconstructed rabbit (rACE2, two lower models) of angiotensin-converting enzyme-2 at pH 5.0 (two left models) and at pH 7.0 (two right models). The surfaces of the models are colored according to the electrostatic potential (negative—red, positive—blue), computed at pH 5.0 or pH 7.0, ionic strength 0.0001 mol/L, and temperature 20 °C, and visualized in the range *kT*/*e* = ±6 J/C (the scale on the right); 1 *kT*/*e* = 25.26 mV at 20 °C or 26.73 mV at 37 °C.

**Figure 6 ijms-25-12393-f006:**
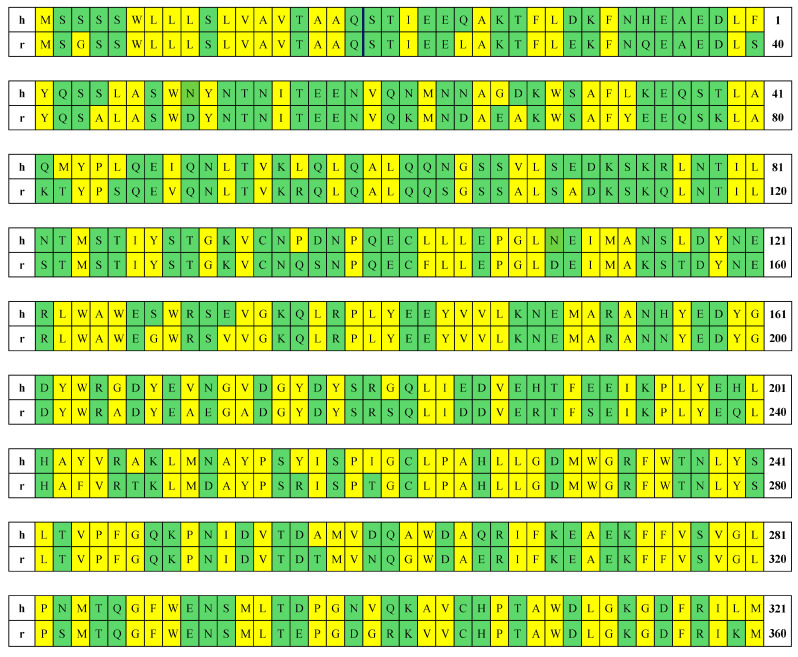
Amino acid sequence of the polypeptide chains of human (rows beginning with **h**) and rabbit (**r**) ACE2. The numbers at the end of every row indicate the first and the last amino acid residue on the corresponding row. The cells of the amino acid residues (denoted by the standard one-letter code) are colored according to their affinity to the water molecules: hydrophilic (green) or hydrophobic (yellow). The vertical lines and the red rectangle denote the beginning and the end of the catalytic domain and the amino acid residues from the zinc-binding motif included in the enzyme active center, respectively.

**Figure 7 ijms-25-12393-f007:**
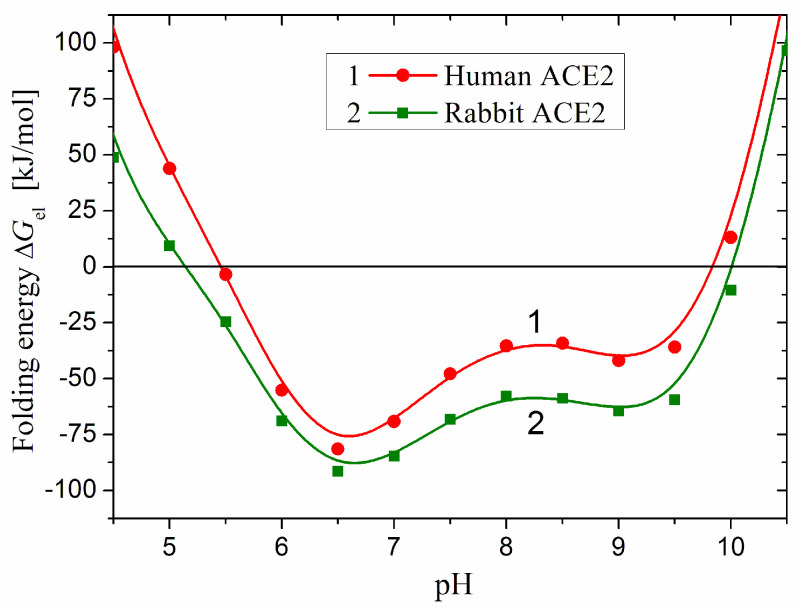
pH dependences of the electrostatic component Δ*G*_el_ of the folding energy Δ*G*_fold_ of the human (hACE2, PDB: 1r42, curve 1) and reconstructed rabbit (rACE2, curve 2) polypeptide chains of angiotensin-converting enzyme-2 at the transformation of the polypeptide chain from fully unfolded (random coil) to folded (globular 3D structure) conformation. The two 3D models are optimized by the program YASARA.

**Figure 8 ijms-25-12393-f008:**
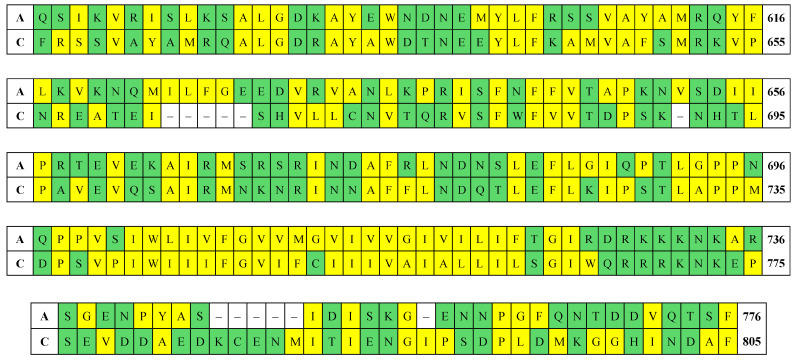
Amino acid sequence of the polypeptide chains of transmembrane collectrin-like segment of the human ACE2 (the rows denoted by **A**) and human collectrin (**C**). The numbers at the end of every row indicate the first and the last amino acid residue on the corresponding row (the numbering corresponds to that in the hACE2 shown in Figure 6). The cells of the amino acid residues (denoted by the standard one-letter code) are colored according to their affinity to water: hydrophilic (green) or hydrophobic (yellow). The absent amino acid residues are denoted by dashes.

**Figure 9 ijms-25-12393-f009:**
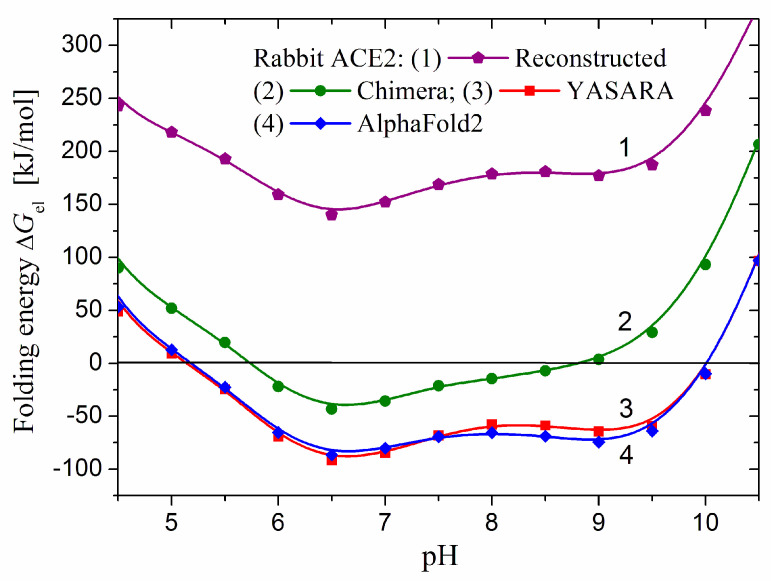
pH dependences of the electrostatic component Δ*G*_el_ of the folding energy of the catalytic domain of rabbit ACE2 polypeptide chain at transformation from random coil to 3D structure for four models: reconstructed on the base of hACE2 (PDB: 1r42, curve 1) and optimized by Chimera (curve 2) or YASARA (curve 3), and created by AlphaFold2 on the base of amino acid sequence of rACE2 (curve 4).

**Figure 10 ijms-25-12393-f010:**
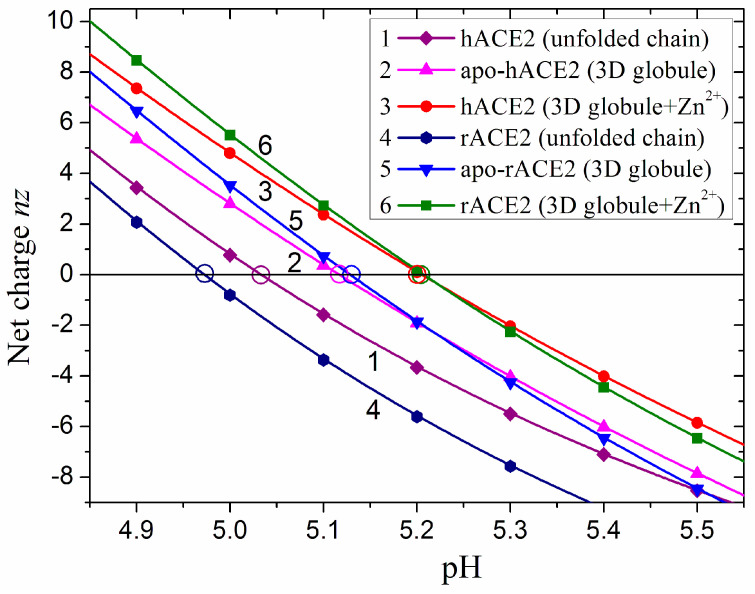
pH dependences of the net electric charge *nz* of the catalytic domain of human (hACE2, curves 1, 2, and 3) and rabbit (rACE2, curves 4, 5, and 6) angiotensin-converting enzyme-2 in aqueous medium when the polypeptide chain is in unfolded conformation (random coil, curves 1 and 4), folded in 3D globule without Zn^2+^ (curves 2 and 5) and when the Zn^2+^ cation is bound in the enzyme active center (curves 3 and 6). The folded 3D conformations correspond to the crystallographic model of hACE2 (PDB:1r42, red curve 3) and to the reconstructed model of rACE2 (green curve 6). The isoelectric points are denoted by open cycles. The net charge *nz* is the algebraic sum of the positive and negative coulombic charges of the polypeptide chain without (curves 1, 2, 4, 5) or with the attached Zn^2+^ cation (curves 3 and 6).

**Figure 11 ijms-25-12393-f011:**
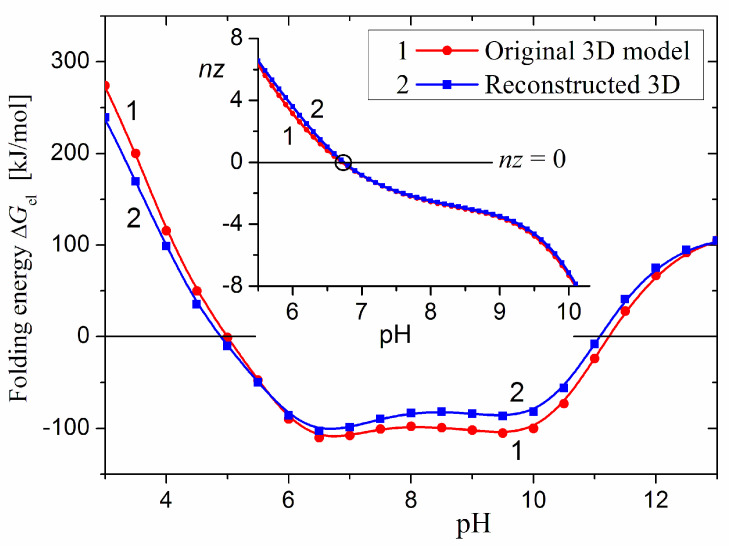
pH dependences of the electrostatic component Δ*G*_el_ of the folding energy (the main figure) and the net electric charge *nz* (the inserted figure) of horse myoglobine according to the original 3D model (PDB: 1AZI, curves 1) and its reconstructed analog (curves 2). The reconstruction of the 3D model of the horse myoglobine is performed on the base of human myoglobine (PDB: 3RGK) considering the difference in the amino acid sequences of the human and horse myoglobine without optimization.

## Data Availability

The original contributions presented in the study are included in the article; further inquiries can be directed to the corresponding author.

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
