# Peer review of "Rabbit and Human Angiotensin-Converting Enzyme-2: Structure and Electric Properties"

_ijms, 2024, doi:10.3390/ijms252212393_

Round 1

Reviewer 1 Report (New Reviewer)

Comments and Suggestions for Authors

The manuscript by Hristova et al. provides a comparative analysis of rabbit and human angiotensin-converting enzyme-2 (ACE2), suggesting rabbits as a potential model for studying coronavirus infection. While the manuscript proposes using rabbits as a model for studying coronavirus infections based on structural similarities in ACE2, it does not fully address the biological relevance of these models, given that SARS-CoV-2 infection in rabbits is typically asymptomatic and requires high viral loads for infection (PMID: 34620166, PMC7832544). This aspect raises significant concerns about the model's ability to faithfully mimic human disease pathology and response to infection. Expanding on these limitations would provide a more balanced view of the potential application of these findings.

Major comments:

1.       The comparative analysis between rabbit and human ACE2 primarily focuses on structural aspects without delving into the functional implications of these differences. The manuscript could be substantially improved by incorporating studies or data on how these structural variations influence ACE2's interaction with the SARS-CoV-2 spike protein, potentially affecting the virus's entry into cells. Insights into binding efficiencies, enzyme kinetics, or substrate specificities could provide valuable information on the suitability of rabbits as a model organism for this virus.

2.       The manuscript is well-presented but does not sufficiently advance the understanding of ACE2's role in SARS-CoV-2 infection beyond what has been previously reported in the literature (e.g., conservation of contact residues in ACE2 critical for binding to the spike protein, PMID: 27578435 "Contact residues of human and rabbit ACE2 critical for binding viral spike (S) glycoprotein are relatively well conserved ). It would benefit from exploring under-researched or novel aspects of ACE2 functionality or its role in disease to enhance its impact and relevance in the field.

3.       The selection of rabbit ACE2 for comparison is not clearly justified in the abstract. Clarifying the specific reasons for this choice, such as particular similarities or differences known to impact viral interaction, would strengthen the rationale for this study.

4.       The year 2000 reference to the ACE enzyme should be supported by citations to ensure the accuracy and reliability of the historical context provided (Page 2, first line).

5.       For original figures such as Figure 1, it is crucial to specify the creation software used. This addition will aid in replicating the study's findings and validate the methods used.

6.       Figures 2 and 6 are described as chaotic. Recommending the authors to present these data in a traditional sequence alignment format would improve clarity and facilitate a better understanding of the critical points of comparison between rabbit and human ACE2.

7.       The discussion of the pH-dependent variation in surface electrostatic potential is intriguing but needs more detail. Specifying how changes in the electrostatic potential influence ACE2's interactions with biological molecules could clarify the functional implications of these findings.

8.       Abstract: The statement mentions "the pH of the medium." Could the authors specify which medium they are referring to? Clarification will help understand the context and relevance of the pH conditions discussed.

Minor Comment:

The absence of line numbers can hinder the review process. Suggesting their reinstatement will facilitate easier referencing during revisions and improve the clarity of subsequent reviews.

Author Response

Reviewer 2 Report (New Reviewer)

Comments and Suggestions for Authors

The manuscript "Rabbit and human angiotensin-converting enzyme-2: Structure and electric properties" by Hristova et al. reports a calculation analysis of the two variants of ACE2. The manuscript's scientific value is very low, and the text is extremely inefficient. I have no issues with the work itself, but more analysis and calculation can be easily made. Moreover, there are more advanced computational tools (e.g. Alphafold) that can be applied, but none of them are discussed in the manuscript.

The abstract contains too detailed information that has a limited impact. The length does not mean importance! The same criticism can be easily applied to the majority of the text. It looks like a chapter in a textbook for undergraduates, not like a scientific paper. E.g., why the Figure 1 is there? None of the things described in the figure are discussed further in the manuscript. Why do the authors mention that residues are numbered from the N-terminus on page 4?

Why are the authors surprised that there is nearly no structural change when their computational approach does not allow movement of the residues per se? At least, one alternative computational approach should be considered (if not applied) in the reviewed manuscript.

Please, consider deletion of the first paragraph in section 2.2 - electric charge.

Please, consider deletion of discussion of properties of unfolded protein. The mission of the whole manuscript is to convince the reader that the rabbit variant is suitable as an animal model. In both cases, folded protein is crucial to maintain the function.

Section 3.2 - Delete or rephrase the first sentence of the second manuscript. Does EACH protein really have so many experimental models in the PDB? If so, why do we need the calculation approaches then?

Please, consider the deletion of properties at total extremes of pH like 1.0 or 14. None of the proteins would be exposed to these environments.

Minor issues:

Page 3 - rephrase the first sentence. It is not clear to what target sequence is the identity calculated.

Page 4 - rephrase the first sentence to: "There is no 3D model of a free rabbit ACE2 in the PDB, ..."

Figure 2 - use small "h" and "r" in the table. All other sections denoting the organisms are in small letters.

Section 2.2, second paragraph - Consider deletion of the last sentence. This should be known from school.

Figure 6 - no important information, already contained in Figure 2, should be deleted.

Page 19 - typo - "substitude"  change to "substitute"

Page 19 - delete the sentence "Finding such a couple ..." This sentence is incorrect and not necessary.

I see no point in having section 3.8 in the manuscript at all.

My final verdict is "Major revision", but I see the rejection also as an option. I believe we should maintain some readability and quality of the scientific text. Wasting the time with reading the textbooks is not appropriate. In addition to that, it is difficult to find important information in such a large manuscript. Nevertheless, I want to encourage the authors to rewrite the manuscript because the information may still be valuable for a limited number of readers. As a target length of the manuscript, I suggest eight pages as a maximum. Comparison with more computational approaches should be also considered.

Round 2

Reviewer 1 Report (New Reviewer)

Comments and Suggestions for Authors

I have no further comments. 

Reviewer 2 Report (New Reviewer)

Comments and Suggestions for Authors

The authors addressed all issues I raised in my first review.

This manuscript is a resubmission of an earlier submission. The following is a list of the peer review reports and author responses from that submission.

Round 1

Reviewer 1 Report

Comments and Suggestions for Authors

Reviewer 2 Report

Comments and Suggestions for Authors

The authors presented a purely computational article comparing the rabbit and human ACE2 protein, including their reconstructed structures, physical/chemical properties, surface charges, etc. To be honest, this manuscript is far from complete. Conclusion is weak and cannot be supported by the data.

1. I don't think this work makes much sense. It is rather a student exercise for computational protein science. The structure prediction, chemical/physical properties can be computed with simple tools within clicks. Even some online tools can do it.

2. The conclusion is contradictory to many common knowledge of COVID-19. The authors concluded that the rabbit ACE2 and human ACE2 are very similar, so that the rabbit can serve as an animal model to investigate COVID infection and extrapolate the conclusions to human. However, it is widely known that the rabbit does not contribute to the spread of COVID-19, and the infectivity of SARS-COV-2 to rabbit is low. This shows that the rabbit ACE2 cannot mimic the properties of human ACE2 during each step in the viral life cycle. Indeed, there's not a single rodent animal model on COVID-19 using the endogenic ACE2. The mouse models use transgenic human ACE2. Therefore, these facts rejected the author's conclusion.

3. The authors did not do any single experiment to validate their computation. In fact, there are many irrational results. For example, Fig.7, the rabbit ACE2 has a high folding energy. The theoretical foldable region (deltaG<0) spans only pH=6~7.2. This means that under common physiological pH=7.4, the rabbit ACE2 cannot fold. This is absolutely irrational and unrealistic. Therefore, the computation is wrong. What's more, there's a huge difference (>50KJ/mol) between the folding energy of hACE2 and rACE2. If the author claims that these two proteins are similar, how can their folding to be so different? If there's so huge a difference between their folding property, how can rACE2 mimic the function of hACE2?

Comments on the Quality of English Language

The English writing is tedious and unclear. Should be intensively proofread.

Round 2

Reviewer 1 Report

Comments and Suggestions for Authors

Thank you for thoroughly addressing my comments and suggestions. I appreciate the detailed responses and improvements made throughout the manuscript, especially in clarifying key aspects such as the rationale for using rabbit models, the electrostatic implications on ACE2 functionality, and the structural comparison of rACE2 and hACE2. The revisions to figures and the incorporation of additional references further enhance the clarity and depth of the work. I look forward to your future articles that will expand on the points discussed.